# Association of Diabetes Mellitus with All-Cause and Cause-Specific Mortality among Patients with Metabolic-Dysfunction-Associated Fatty Liver Disease: A Longitudinal Cohort Study

**DOI:** 10.3390/jpm13030554

**Published:** 2023-03-20

**Authors:** Yixuan Zhu, Chuan Liu, Xiaoming Xu, Xiaoyan Ma, Jiacheng Liu, Zhiyi Zhang, Fuchao Li, Danny Ka-Ho Wong, Zhiwen Fan, Chao Wu, Xiaolong Qi, Jie Li

**Affiliations:** 1Department of Infectious Diseases, Nanjing Drum Tower Hospital, Clinical College of Nanjing Medical University, Nanjing 210008, China; 2Center of Portal Hypertension, Department of Radiology, Zhongda Hospital, Medical School, Southeast University, Nanjing 210009, China; 3Department of Infectious Diseases, Nanjing Drum Tower Hospital Clinical College of Traditional Chinese and Western Medicine, Nanjing University of Chinese Medicine, Nanjing 210023, China; 4Department of Infectious Diseases, The Affiliated Drum Tower Hospital of Medical School of Nanjing University, Nanjing 210008, China; 5Department of Gerontology, The Affiliated Drum Tower Hospital of Medical School of Nanjing University, Nanjing 210008, China; 6Department of Medicine, The University of Hong Kong, Hong Kong; 7State Key Laboratory of Liver Research, The University of Hong Kong, Hong Kong; 8Department of Pathology, The Affiliated Drum Tower Hospital of Medical School of Nanjing University, Nanjing 210008, China; 9Institute of Viruses and Infectious Diseases, Nanjing University, Nanjing 210093, China

**Keywords:** MAFLD, diabetes mellitus, mortality, cardiovascular-related mortality, cancer-related mortality

## Abstract

Background: Diabetes mellitus (DM) is a comorbidity commonly presenting with metabolic-dysfunction-associated fatty liver disease (MAFLD); however, few tests for interaction have been reported. Our target was to evaluate the prognostic implications of DM in patients with different forms of MAFLD. Methods: Using data from the Third National Health and Nutrition Examination Survey (NHANES III) in the United States, we screened 14,797 participants aged 20–74 who received ultrasound examinations from 1988–1994. Among them, 4599 patients met the diagnosis of MAFLD, and we defined mortality as the outcome event. Survival analysis of competitive risk events was performed using Cox regression and sub-distributed risk ratio (SHR). Results: During 21.1 years of follow-up, cardiovascular diseases seemed to be the most common cause of death among MAFLD patients. Of them, DM was present in 25.48% and was independently associated with increased risk of all-cause mortality (HRs: 1.427, 95% CIs: 1.256–1.621, p < 0.001) and cause-specific mortality (cardiovascular-related mortality (HRs: 1.458, 95% CIs: 1.117–1.902, p = 0.005), non-cardiovascular-related mortality (HRs: 1.423, 95% CIs: 1.229–1.647, p < 0.001), and non-cancer-related mortality (HRs: 1.584, 95% CIs: 1.368–1.835, p < 0.001), respectively). Surprisingly, this association was more significant for young patients (p-value for interaction <0.001). Moreover, DM had a greater risk of all-cause and cause-specific mortality among overweight and obese MAFLD patients (p-value for interaction <0.001). Conclusions: DM increased the risk of all-cause and cause-specific mortality (cardiovascular-related, non-cardiovascular-related, and non-cancer-related) in MAFLD patients, especially in younger patients with excess obesity.

## 1. Introduction

Metabolic-dysfunction-associated fatty liver disease (MAFLD), once known as non-alcoholic fatty liver disease (NAFLD), has emerged as the most prevalent liver disease in the world, impacting over 30% of the general population [1,2]. MAFLD includes a spectrum of liver histological progressions ranging from simple steatosis to varying degrees of fibrosis with infiltration of inflammatory cells. Compared with NAFLD patients, MAFLD patients are more at risk of having broad metabolic traits and higher fibrosis levels [3,4]. Like other chronic liver diseases, it can develop into cirrhosis if fibrosis progresses [5]. In addition, hepatocellular carcinoma (HCC) may happen irrespective of any significant fibrosis or cirrhosis [6].

Although hepatic steatosis progresses slowly in the short term, patients with MAFLD often have metabolic diseases such as hypertension, hyperlipidemia, obesity, and diabetes mellitus (DM) [7]. There were 32 million diabetics aged 20–79 in the United States in 2021, and it as is estimated that 4 million people were undiagnosed, the number of patients with fatty liver complicated with diabetes may be more than reported [8]. Numerous studies have shown a strong association between DM and NAFLD [9]. A 2020 meta-analysis of 22 studies from 16 cohorts showed that DM tripled the risk of severe liver disease, while other metabolic factors were less hazardous than DM [10]. DM seems to be the most powerful predictor of adverse clinical outcomes among these comorbidities [11,12,13]. Compared with NAFLD patients, MAFLD patients had a higher risk of diabetes comorbidities due to more glucose metabolism-related disorders [14]. However, studies on the long-term prognosis of patients classified as MAFLD are minimal. The association between DM and mortality risk from various causes and specific causes in different forms of MAFLD patients is unclear.

Our study used extensive, single, population-based data from the Third National Health and Nutrition Examination Survey (NHANES III) and characterized participants based on the MAFLD definition. We aimed to identify DM with all causes and specific causes of mortality in MAFLD patients.

## 2. Materials and Methods

### 2.1. Data Collection and Study Setting

We used data from the Third National Health and Nutrition Examination Survey (NHANES III), which was conducted by the National Center for Health Statistics (NCHS) of the Centers for Disease Control (CDC) of the United States. All surveys used a complex, multi-stage, stratified subset of non-institutional civilian participants from 1988 to 1994. The NCHS of the CDC provided a link to mortality data, as of 31 December 2015, from the nationwide death index. The study was based on the Declaration of Helsinki. The NHANES protocol was approved by the National Center for Health Statistics Research Ethics Review Board.

### 2.2. Study Variable Measurement

The ultrasound examinations were performed on 14,797 adults aged 20–74 years who were examined in NHANES III during 1988–1994. This study used a DVD-VHS videocassette recorder system to digitize videotapes of ultrasounds, and a board-certified radiologist trained three ultrasound readers to evaluate hepatic steatosis. Excluding missing and ungradable images, the following five ultrasonographic findings were used to evaluate the extent of steatosis: liver-to-kidney contrast, parenchymal brightening, deep beam attenuation, vessel walls, and gallbladder wall. Professionals classified hepatic steatosis into ordinary, mild, moderate, and severe. DM may be diagnosed if any of the following conditions are met: (1) self-reported medical history of DM, (2) fasting plasma glucose (FPG) ≥ 126 mg/dL (7.0 mmol/L), (3) 2 h plasma glucose (2 h PG) ≥ 200 mg/dL (11.1 mmol/L) during OGTT, and (4) HbA1c ≥ 6.5% [15]. Several new diagnostic criteria have been proposed for MAFLD based on evidence of hepatic steatosis (detected either by imaging techniques, blood biomarkers/scores, or by liver histology) as well as at least one of the following three conditions: overweight/obesity (BMI ≥ 25 kg/m^2^), the presence of DM, and lean/normal body weight with signs of metabolic dysregulation [16]. Finally, 4599 MAFLD patients were screened out.

### 2.3. Clinical and Laboratory Assessments

We obtained the following data from the dataset: demographic indicators regarding age, gender, ethnicity, BMI, and poverty income ratio (PIR). The definition of hypertension is a blood pressure value ≥ 140/90 mmHg or current antihypertensive medication use [17]. The serological parameters retrieved from the database and studied included alanine aminotransferase (ALT), aspartate aminotransferase (AST), C-reactive protein (CRP), alkaline phosphatase (ALP), total cholesterol (TC), high-density lipoprotein cholesterol (HDL-C), total triglyceride (TG), serum ferritin (SF), vitamin A, vitamin C, vitamin E, serum selenium, platelet (PLT), uric acid (UA), total protein, albumin (ALB), serum creatinine (SCR), HOMA-IR, and estimated glomerular filtration rate (eGFR). The presence of hepatic fibrosis was assessed using the fibrosis score 4 (FIB-4) and non-invasive NAFLD fibrosis score (NFS), as these scores may provide the best diagnostic yield for detecting advanced fibrosis [18]. The specific calculation methods of the above formulas have been elaborated in previous studies and will not be repeated here [19,20,21,22].

## 3. Results

### 3.1. Baseline Characteristics of the Included Population

Figure 1 depicts the patient characteristics in the current study. We screened 14,797 participants between 20 and 74, including 4599 patients with MAFLD. Of them, 25.5% of patients were diagnosed with DM. Compared to the non-DM group, the DM patients were older (median age: 60 vs. 42, *p* < 0.001) and more frequently female (55.6% vs. 48.8%, *p* < 0.001) (Appendix A).

PSM yielded 1172 pairs of DM-MAFLD and non-DM MAFLD patients with balanced characteristics (age and gender). Table 1 shows that DM was less common in non-Hispanic Whites (33.3%) but more common in Mexican Americans (39.2%, *p* < 0.001). Patients with DM were more commonly obese (54.9% vs. 42.2%, *p* < 0.001) and had a lower poverty income ratio (PIR < median, 56.2% vs. 43.7%, *p* < 0.001). In terms of serological indicators, DM patients had higher levels of liver enzymes (ATL: 18 (13, 27) vs. 16 (12, 24), ALP: 95 (77, 117) vs. 86 (71, 103), *p* < 0.001), a worse metabolic profile (TG: 2.07 (1.5, 3.08) vs. 1.64 (1.15, 2.36), HDL-C: 1.1 (0.93, 1.34) vs. 1.22 (1.01, 1.45), *p* < 0.001), and poorer liver function (ALB: 41 (38, 43) vs. 41 (39, 44), PLT: 263 (220, 315.5) vs. 271 (228, 320.5), *p* < 0.05). Moreover, DM patients showed more severe hepatic steatosis (27.8% vs. 18.3%, *p* < 0.001) and higher noninvasive liver fibrosis scores (NFS ≥ 0.676, 20.2% vs. 8.1%; FIB-4 ≥ 2.67, 4.4% vs. 3.6%, *p* < 0.001).

### 3.2. Relationship between DM and Overall and Specific Causes of Mortality in MAFLD Patients

With death as the outcome, age, gender, ethnicity, PIR, hypertension, DM, vitamin C, ALP, ALB, SCR, and NFS scores were finally selected for further adjustment using LASSO regulation. By univariate as well as multifactorial Cox regression, our study confirmed DM as an independent risk factor for death in MAFLD patients (Appendix A). There were 1366 reported deaths during the median 21.1-year follow-up period. Regarding cause-specific mortality, cardiovascular disease accounted for 308 deaths, and cancer accounted for 302 deaths. The most common cause of death in MAFLD patients appeared to be cardiovascular disease (Figure 2A).

In patients with MAFLD, DM was associated with a higher risk of all-cause mortality (Figure 2B, HRs: 1.459, 95% CIs: 1.31–1.623). The conclusion persisted in the full age group and across BMI levels (Appendix A). Relative to non-DM patients, DM patients had a higher risk of cardiovascular-related mortality (Figure 3A, HRs: 1.475, 95% CIs: 1.176–1.849), non-cardiovascular-related mortality (Figure 3B, HRs: 1.446, 95% CIs: 1.280–1.634), and non-cancer-related mortality (Figure 3C, HRs: 1.622, 95% CIs: 1.435–1.834) (Table 2). However, we found no differences in cancer-related mortality between DM and non-DM patients (Figure 3D, Table 2).

After considering known demographic variables (model 1: adjusted by age, gender, ethnicity, PIR) and traditional risk factors (model 2: model 1+ adjusted by hypertension and BMI), the association of DM with mortality for all causes and specific causes remained significantly higher (Table 2). In fully adjusted models (model 3: model 2 + adjusted by vitamin C, ALP, ALB, SCR, and NFS scores), those with diabetes had a 42.7% (HRs: 1.427, 95% CIs: 1.256–1.621, *p* < 0.001), 45.8% (HRs: 1.458, 95% CIs: 1.117–1.902, *p* = 0.005), 42.3% (HRs: 1.423, 95% CIs: 1.229–1.647, *p* < 0.001), and 58.4% (HRs: 1.584, 95% CIs: 1.368–1.835, *p* < 0.001) higher risk of all-cause mortality, cardiovascular mortality, non-cardiovascular-related mortality, and non-cancer-related mortality, respectively, than patients DM. Similarly, the association of DM with cancer-related mortality remained nonsignificant after adjustment for covariates in all models (Table 2).

### 3.3. Subgroup Analysis of the Association of DM with Mortality in MAFLD Patients

In the stratified analyses, the risks of DM for all-cause mortality differed across subgroups of age, gender, ethnicity, PIR, hypertension, BMI, NFS scores, and severities of hepatic steatosis (*p*-value for interaction < 0.05). The DM group had higher risks of all-cause mortality in females (HRs: 1.641 (1.393, 1.933) vs. 1.532 (1.305, 1.798)) and Mexican-American patients (HRs:1.724 (1.387, 2.142) vs. 1.502 (1.269, 1.778) vs. 1.416 (1.110, 1.807)) with middle income (1 ≤ PIR < median, HRs: 1.726 (1.387, 2.142) vs. 1.546 (1.222, 1.957 vs. 1.400 (1.184, 1.655)). We also identified a remarkable variation in the effect of DM on mortality at different baseline ages (*p*-value for interaction < 0.001), and this correlation was much stronger in patients under 45 years of age than in those over 59 years (HRs: 2.258 (1.435, 3.551) vs. 1.415 (1.236, 1.620)). When we classified cohorts according to BMI (*p*-value for interaction < 0.001), DM was associated with higher all-cause mortality in overweight (25 ≤ BMI < 30 kg/m^2^, HRs: 1.675, 95% CIs: 1.390–2.017, *p* < 0.001) and obese (BMI ≥ 30 kg/m^2^, HRs: 1.629, 95% CIs: 1.372–1.935, *p* < 0.001) patients with MAFLD, while this association was not demonstrated in normal and lean patients (BMI < 25 kg/m^2^, *p* = 0.273). With the increasing NFS scores, the mortality risk of DM decreased (NFS ≥ 0.676, *p* = 0.346) (Figure 4). When restricted to cause-specific mortality, we observed similar patterns of results in subgroups with age, BMI, and NFS scores. Among patients with hypertension (*p* = 0.132) and severe hepatic steatosis (*p* = 0.334), the association between DM and cardiovascular mortality turned null (Figure 5). The association between DM and cancer-related mortality remained insignificant in all subgroups except in patients with severe liver steatosis (HRs: 1.63, 95% CIs: 1.008–2.637, *p* = 0.046) (Appendix A). Death outcomes from non-cancer-related mortality (Figure 6) and non-cardiovascular-related mortality (Appendix A) were similar to those for all-cause mortality.

## 4. Discussion

This large prospective study of U.S. adults aged 20–74 years evaluated mortality outcomes in individuals with MAFLD. First, our analysis indicated that DM increased mortality risk by 40–60% in patients with MAFLD while having no significant effect on cancer-related mortality. Especially in younger patients (individuals aged < 45 years) with excess weight (BMI ≥ 25 kg/m^2^), DM showed a more significant relationship with increased mortality risk. Second, cardiovascular diseases seemed to be the most common cause of death among MAFLD patients. Furthermore, compared to non-diabetic groups, diabetics were older, mostly female, and had a higher metabolic burden on the hepatic system. Overall, DM patients were more likely to go on to “severe MAFLD” [23].

We confirmed that DM was an independent risk factor for death among patients with MAFLD, increasing the risk of all-cause mortality by 42.7%. DM often co-exists with MAFLD, and an intimate association has been demonstrated [24]. The link between DM and fatty liver disease is more complex than previously appreciated and appears to be bidirectional [25]. MAFLD increases the risk for the development of DM, and DM is an aggravating factor for MAFLD. A common potential risk factor is insulin resistance, resulting in the accumulation of steatosis [26]. Our study expands the results of other studies. Similar to what has been indicated in studies of NAFLD, devastating macrovascular complications and microvascular complications of DM, such as coronary artery disease and diabetic nephropathy, can significantly increase all-cause mortality in patients with MAFLD, especially cardiovascular-related mortality [27]. Considering the strong association between MAFLD and cardiovascular risk factors including abdominal obesity, hypertension, atherogenic dyslipidemia, and insulin resistance, the result is not surprising [28]. Therefore, the clinical emphasis should shift from MAFLD as a single-organ entity to a multisystemic disease [29].

Epidemiological data suggest that DM increases morbidity and mortality from many cancers [30]. However, the results may differ depending on which denominator population (MAFLD or whole population) is used. Although our study did not find an independent association between DM and cancer-related mortality in all MAFLD patients, we still acknowledge that the presence of DM promotes the development of advanced liver disease [31,32,33]. In addition, we found that with increasing NFS fibrosis scores, the association of diabetes with mortality decreases. In our study, the fibrosis stage was found to be the strongest predictor of hepatocellular carcinoma, while the role of DM appears to be overshadowed due to competing risks [34]. Other metabolic factors such as obesity have also been proven to be risk factors for cancer development [35]. However, site-specific cancer mortality data are not available, and more research is needed to examine the relationship between DM (or other metabolic factors) and MAFLD-related cancer risk.

In addition to the strong association between DM and mortality, we also found that DM’s impact on all cause-related deaths is more pronounced in female and Mexican-American patients with middle income. Many lifestyle and environmental factors may affect blood glucose and mortality risk in a gender-specific manner. Low education, occupation, and income may largely contribute to unhealthy lifestyle behaviors and are therefore related to a high risk of obesity, especially in female patients [36]. In addition, regarding racial differences in DM, our findings were similar to those of previous epidemiological studies, with Mexican Americans having poor glycemic control and a higher incidence of diabetes-related complications [37]. Comparable data from low- and middle-income patients are indeed lacking, but similar findings have been confirmed in high-income patients [38]. The effect of DM on mortality in high-income patients may be influenced by improved food availability (including fat composition, fresh fruits, and vegetables), pharmacotherapy, and better prevention strategies [39]. Concurrent comparison of ethnicity, gender, and socioeconomic discrepancies in parallel may better account for the impact of DM on specific populations such as MAFLD, which we believe is critical for targeted public-health decision making.

Traditionally, DM has been regarded as a disease of middle-aged and elderly individuals, while the situation in young patients seems even less optimistic. In an epidemiologic analysis from 1985 to 2015 in the USA, the rate of all-cause mortality due to diabetes was reported to be decreasing. However, this positive pattern was not observed in younger adults (20 to 44) [40]. Our findings demonstrated that younger age at DM diagnosis was associated with more significant mortality [41,42]. Similarly, among patients with NASH, a meta-analysis of 80 studies from 20 countries suggested that patients with DM are much younger [9]. The progression of DM to liver inflammation may start at a young age and follow a progressive course. Regarding the mechanisms, researchers found more rapid deterioration in β-cell function in those with early-onset DM [43]. Additionally, in our study, young patients with DM had a higher BMI level than older patients, which is consistent with the previous studies [44], suggesting that obesity-related mechanisms may also be a key player. Being one of the three defining criteria for MAFLD, excess weight is as critical a determinant of adverse clinical outcomes such as DM [45,46]. However, the evidence regarding the relationship between BMI and mortality remains inconclusive. A 2016 meta-analysis of 21 studies suggested a non-linear relationship between the two in the diabetic population, with the lowest risk of approximately 33 kg/m^2^ [47]. However, we have not found this obesity paradox in our study, and DM significantly correlated with the mortality risk in MAFLD patients suffering from overweight and obesity. The results emphasized the value of managing blood glucose and weight while controlling the progression of hepatic steatosis. Special attention to preventing or screening for DM in younger MAFLD patients could be particularly beneficial.

Despite the large sample size, long follow-up period, and comprehensive analysis of the role of DM mortality outcomes in general and different types of MAFLD, it is important to note the following limitations. First, the baseline data for this cohort were derived from 1988–1994, and the prevalence of DM in patients with MAFLD might not reflect its contemporary extent. Second, all covariates were available only at baseline, so we could not capture changes in possible confounders over time, including glycemic control, during follow-up. Third, due to the restricted nature of these data, there were no follow-up data on liver-related events, which prevented us from distinguishing between liver-related and all-cause mortality. Additional studies should investigate whether the simultaneous presence of MAFLD and DM synergistically increases the risk of liver-related adverse outcomes.

## 5. Conclusions

In conclusion, DM increased the risks of all-cause, cardiovascular-related, non-cardiovascular-related, and non-cancer-related mortality by 40–60% among patients with MAFLD. Especially in younger patients with excess weight, DM showed a more significant relationship with increased mortality risk.

## Figures and Tables

**Figure 1 jpm-13-00554-f001:**
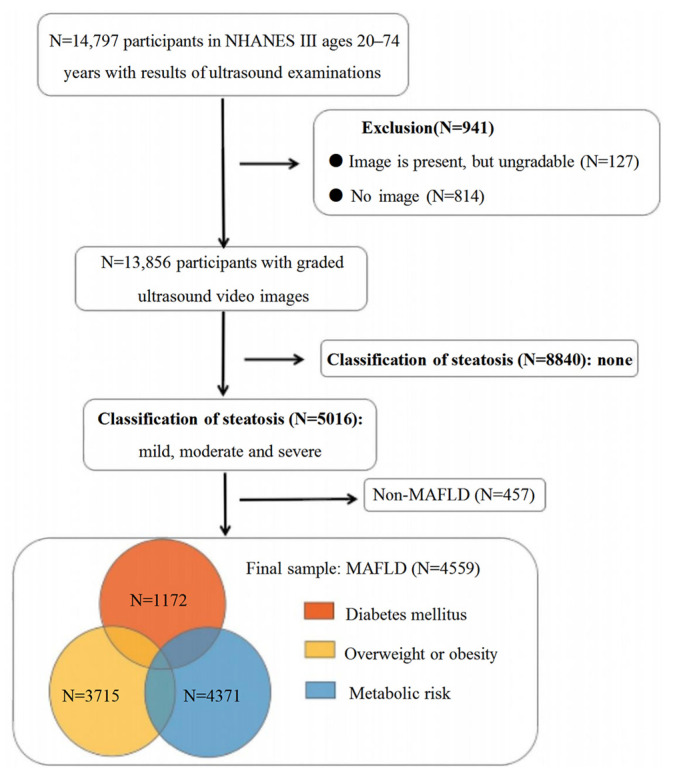
Flow-chart of the study participants.

**Figure 2 jpm-13-00554-f002:**
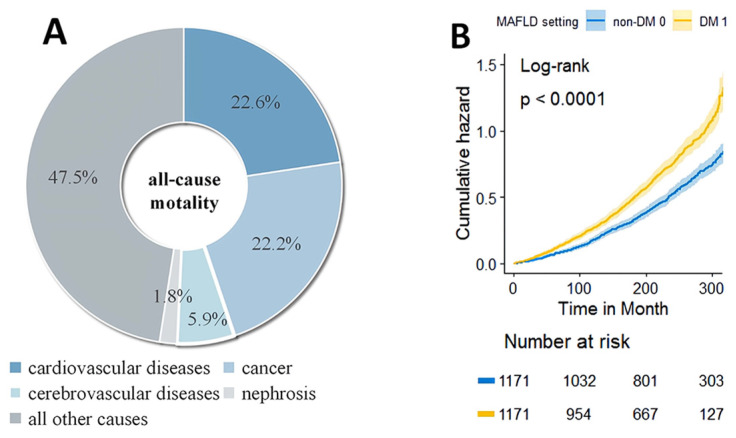
Composition and survival analysis of all-cause mortality. (**A**) Leading cause of death among all-cause mortality. (**B**) Kaplan–Meier estimation of all-cause mortality.

**Figure 3 jpm-13-00554-f003:**
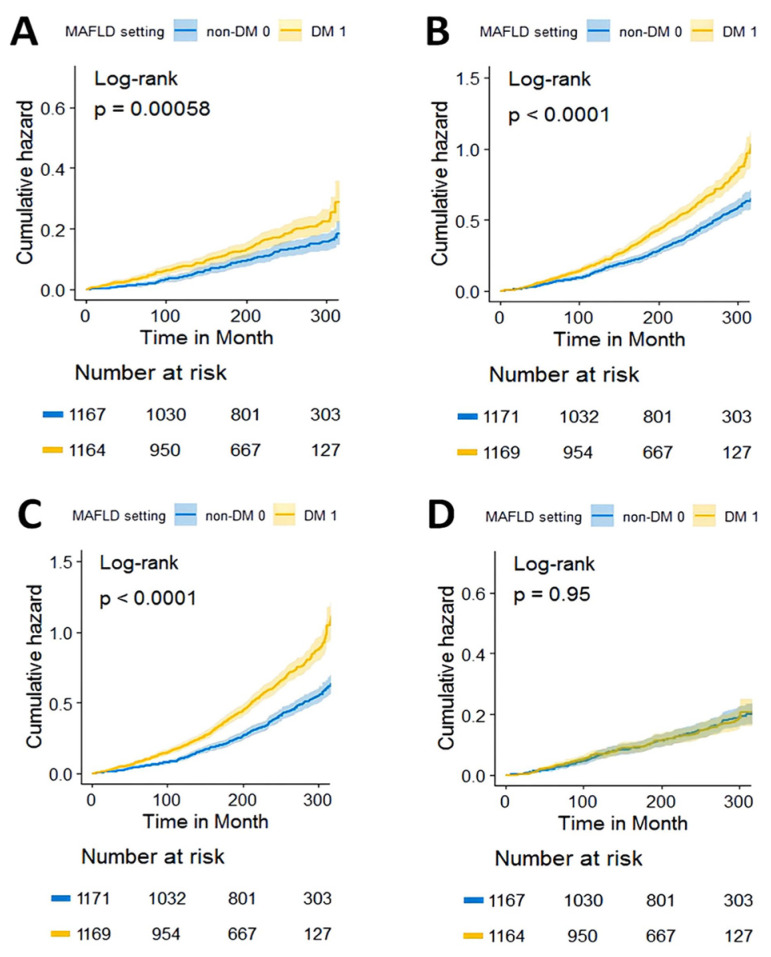
Kaplan–Meier estimations of cause-specific mortality. (**A**) Cardiovascular-related mortality, (**B**) non-cardiovascular-related mortality, (**C**) non-cancer-related mortality, and (**D**) cancer-related mortality.

**Figure 4 jpm-13-00554-f004:**
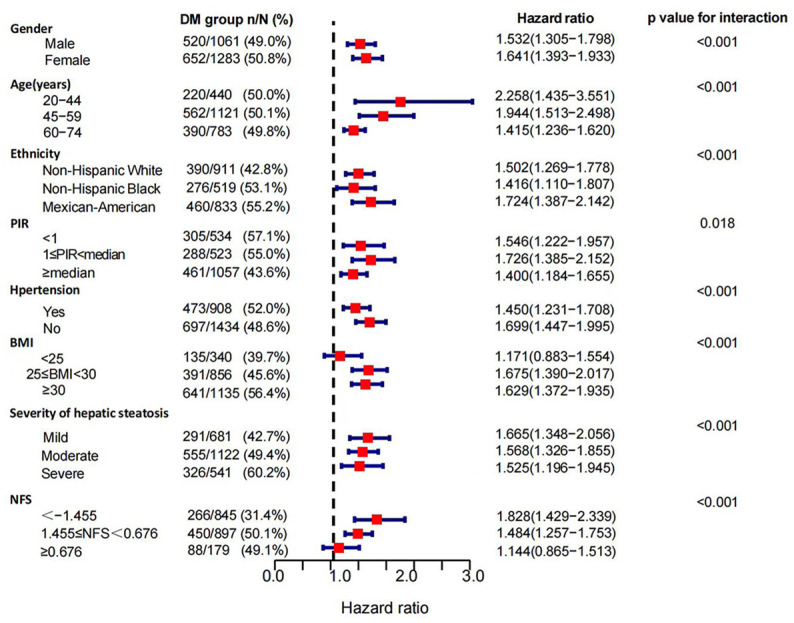
Subgroup analysis of all-cause mortality in MAFLD patients. Patients with diabetes compared to non-diabetics. The association between DM and mortality risk from all causes was stratified by baseline age, gender, ethnicity, PIR (poverty to income ratio), hypertension, BMI, the severity of hepatic steatosis, and NFS scores in MAFLD patients. Models were adjusted for age (not for the age subgroup), gender (not for the gender subgroup), ethnicity (not for the ethnicity subgroup), PIR (not for the PIR subgroup), hypertension (not for the hypertension subgroup), and BMI (not for the BMI subgroup).

**Figure 5 jpm-13-00554-f005:**
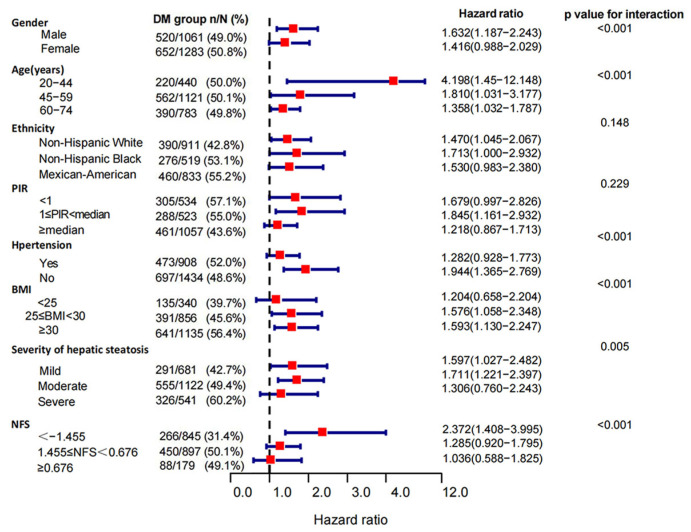
Subgroup analysis of cardiovascular mortality in MAFLD patients. Patients with diabetes compared to non-diabetics. The association between DM and mortality risk from cardiovascular causes was stratified by baseline age, gender, ethnicity, PIR (poverty to income ratio), hypertension, BMI, the severity of hepatic steatosis, and NFS scores in MAFLD patients. Models were adjusted for age (not for the age subgroup), gender (not for the gender subgroup), ethnicity (not for the ethnicity subgroup), PIR (not for the PIR subgroup), hypertension (not for the hypertension subgroup), and BMI (not for the BMI subgroup).

**Figure 6 jpm-13-00554-f006:**
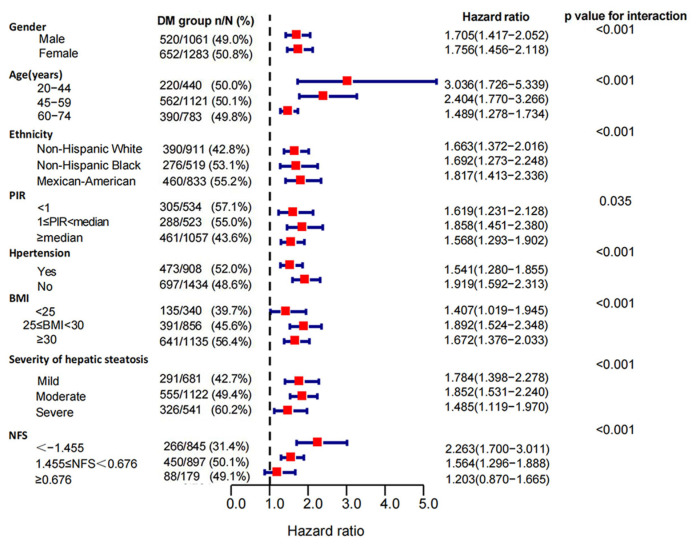
Subgroup analysis of non-cancer-related mortality in MAFLD patients. Patients with diabetes compared to non-diabetics. The association between DM and mortality risk from non-cancer related causes was stratified by baseline age, gender, ethnicity, PIR (poverty to income ratio), hypertension, BMI, the severity of hepatic steatosis, and NFS scores in MAFLD patients. Models were adjusted for age (not for the age subgroup), gender (not for the gender subgroup), ethnicity (not for the ethnicity subgroup), PIR (not for the PIR subgroup), hypertension (not for the hypertension subgroup), and BMI (not for the BMI subgroup).

**Table 1 jpm-13-00554-t001:** Demographics and clinical characteristics in MAFLD patients stratified by diabetes status.

	Overall Cohort (*n* = 2344)	Non-DM Cohort (*n* = 1172)	DM Cohort (*n* = 1172)	*p*-Value
Male, gender (%)	1061 (45.3)	541 (46.2)	520 (44.4)	0.384
Age, years.	60 (48.67)	60 (48.67)	60 (48.67)	0.995
Ethnicity (%)				<0.001
Non-Hispanic White	911 (38.9)	521 (44.5)	390 (33.3)	
Non-Hispanic Black	519 (22.1)	243 (20.7)	276 (23.5)	
Mexican American	833 (35.5)	373 (31.9)	460 (39.2)	
Other	81 (3.5)	35 (3.0)	46 (3.9)	
PIR				<0.001
<1	533 (25.2)	228 (21.5)	305 (28.9)	
1 ≤ PIR < median	523 (24.7)	235 (22.2)	288 (27.3)	
≥median	1058 (50.0)	597 (56.3)	461 (43.7)	
Hypertension (%)	908 (38.8)	435 (37.1)	473 (40.4)	0.1
BMI				<0.001
<25	341 (14.6)	206 (17.7)	135 (11.6)	
25 ≤ BMI < 30	856 (36.7)	465 (39.9)	391 (33.5)	
≥30	1134 (48.6)	493 (42.4)	641 (54.9)	
ALT (U/L)	16 (12, 24)	15 (11, 21)	18 (13, 27)	<0.001
AST (U/L)	20 (17, 27)	20 (17, 26)	20 (17, 27)	0.45
ALP (U/L)	90 (74, 110)	86 (71, 103)	95 (77, 117)	<0.001
CRP (mg/dL)	0.3 (0.2, 0.7)	0.2 (0.2, 0.6)	0.4 (0.2, 0.9)	<0.001
FRP (ng/mL)	127 (65, 243)	110 (55, 196)	153 (77, 285)	<0.001
Vitamin A (umol/L)	2.0 (1.7, 2.4)	2.1 (1.8, 2.5)	2.0 (1.6, 2.4)	<0.001
Vitamin C (umol/L)	36.9 (18.2, 53.4)	38.0 (17.0, 55.6)	35.8 (18.7, 51.7)	0.105
Vitamin E (umol/L)	26.8 (21.6, 33.6)	26.5 (21.5, 32.6)	27.17 (21.6, 35.0)	0.011
Serum selenium (nmol/L)	1.6 (1.4, 1.7)	1.6 (1.4, 1.7)	1.6 (1.5, 1.7)	<0.001
PLT (* 10^9^/L)	267.0 (224.5, 318.3)	271.0 (228.0, 320.5)	263.0 (220.0, 315.5)	0.023
TC (mmol/L)	5.6 (4.9, 6.4)	5.6 (4.9, 6.27)	5.6 (4.9, 6.5)	0.078
TG (mmol/L)	1.9 (1.3, 2.7)	1.6 (1.15, 2.36)	2.1 (1.5, 3.1)	<0.001
HDL-C (mmol/L)	1.2 (1.0, 1.4)	1.2 (1.0, 1.5)	1.1 (0.9, 1.3)	<0.001
Total protein (g/L)	74 (71, 77)	73 (71, 77)	75 (72, 78)	<0.001
Albumin (g/L)	41 (39, 43)	41 (39, 44)	41 (38, 43)	<0.001
UA (umol/L)	333.1 (273.6, 392.6)	339 (284.0, 398.5)	327.1 (261.7, 392.6)	0.001
Scr (mg/dL)	1.0 (0.9, 1.2)	1.1 (0.9, 1.2)	1.0 (0.9, 1.2)	0.008
eGFR category (%)				<0.001
≥90 (Stage 1 CKD)	337 (14.9)	135 (12.1)	202 (17.7)	
60 ≤ eGFR < 90 (Stage 2 CKD)	1349 (59.8)	693 (62.0)	656 (57.6)	
30 ≤ eGFR < 60 (Stage 3 CKD)	550 (24.4)	283 (25.3)	267 (23.4)	
<30 (Stage 4–5 CKD)	20 (0.9)	6 (0.5)	14 (1.2)	
FIB-4 category (%)				0.042
<1.3	1394 (62.7)	666 (60.7)	728 (64.6)	
1.3 ≤ FIB-4 < 2.67	742 (33.3)	393 (35.8)	349 (31.0)	
≥2.67	89 (4.0)	39 (3.6)	50 (4.4)	
NFS category (%)				<0.001
<−1.455	846 (37.5)	580 (51.9)	266 (23.4)	
−1.455 ≤ NFS < 0.676	1090 (48.3)	447 (40.0)	643 (56.5)	
≥0.676	321 (14.2)	91 (8.1)	230 (20.2)	
Severity of hepatic steatosis (%)				<0.001
Mild	681 (29.1)	390 (33.3)	291 (24.8)	
Moderate	1122 (47.9)	567 (48.4)	555 (47.4)	
Severe	541 (23.1)	215 (18.3)	326 (27.8)	0.384

* Continuous values are presented as median (interquartile range) and categorical variables as count (percentage). DM, diabetes mellitus; Non-DM, patients without diabetes mellitus; PIR, poverty income ratio; BMI, body mass index; ALT, alanine transaminase; AST, aspartate transaminase; ALP, alkaline phosphatase; PLT, serum selenium, platelet; TC, total cholesterol; TG, total triglyceride; HDL-C, high-density lipoprotein cholesterol; ALB, total protein, albumin; UA, uric acid; Scr, serum creatinine; eGFR, estimated glomerular filtration rate; NFS, NAFLD fibrosis score; FIB-4, Fibrosis-4 index.

**Table 2 jpm-13-00554-t002:** Cox regression models for the relationship between DM and mortality in MAFLD patients.

		Model 1		Model 2		Model 3	
	Unadjusted HR	HR (95% Cl)	*p*-Value	HR (95% Cl)	*p*-Value	HR (95% Cl)	*p*-Value
All-cause mortality	1.459 (1.310, 1.623)	1.546 (1.381, 1.732)	<0.001	1.564 (1.395, 1.753)	<0.001	1.428 (1.257, 1.623)	<0.001
Cardiovascular mortality	1.482 (1.183, 1.858)	1.546 (1.221, 2.230)	<0.001	1.526 (1.204, 1.935)	<0.001	1.464 (1.122, 1.909)	0.005
Cancer mortality	0.993 (0.791, 1.246)	1.143 (0.903, 1.447)	0.267	1.173 (0.924, 1.489)	0.189	1.026 (0.787, 1.336)	0.851
Non-cardiovascular-related mortality	1.448 (1.282, 1.637)	1.546 (1.358, 1.760)	<0.001	1.575 (1.382, 1.796)	<0.001	1.423 (1.229, 1.647)	<0.001
Non-cancer-related mortality	1.627 (1.439, 1.840)	1.696 (1.489, 1.932)	<0.001	1.709 (1.499, 1.948)	<0.001	1.587 (1.370, 1.838)	<0.001

The results were obtained with Cox proportional hazards analysis and are given as HR with 95% Cl. Model 1: adjusted by age, gender, ethnicity, and poverty income ratio (PIR). Model 2: model 1 + adjusted by hypertension and BMI. Model 3: model 2 + adjusted by vitamin C, ALP, ALB, SCR, and NFS scores.

## Data Availability

The datasets presented in this study are available in online repositories (https://www.cdc.gov/nchs/nhanes/index.htm (accessed on 1 May 2022).

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
