# Peer review of "Association of Diabetes Mellitus with All-Cause and Cause-Specific Mortality among Patients with Metabolic-Dysfunction-Associated Fatty Liver Disease: A Longitudinal Cohort Study"

_jpm, 2023, doi:10.3390/jpm13030554_

Round 1

Reviewer 1 Report

I thank the authors for exploring an important area of diabetes research. 

Authors need to consider the following before it can be published.

Abstract

Please mention the country, study period, participants’ age group and sample size in the “Methods”.

Introduction

The authors need to justify the research in the context of the USA.

Materials and Methods

Please provide details on the study period, survey participants’ age group and total sample size of the survey.

Others

The figures can hardly be seen in the manuscript. Please take care of these.

Author Response

Reviewer: 1 Comments for Transmission to the Authors

I thank the authors for exploring an important area of diabetes research.

Authors need to consider the following before it can be published.

Abstract: Please mention the country, study period, participants’ age group and sample size in the “Methods”.

Authors’ Response: We agree with the reviewer and have supplemented this part on page 1, line 44-47.

Introduction: The authors need to justify the research in the context of the USA.

Authors’ Response: We agree with the reviewer and have added this discussion on page 2, line 76-78.

Materials and Methods: Please provide details on the study period, survey participants’ age group and total sample size of the survey.

Authors’ Response: We agree with the reviewer and have supplemented this part on page 3, line 103-106, 117-118.

Others: The figures can hardly be seen in the manuscript. Please take care of these.

Authors’ Response: Thank you for your comment. We resized the figures and increased the resolution to 600dpi. We are very sorry for the inconvenience.

Reviewer 2 Report

the manuscript was acceptable in its present forum. 

Author Response

Reviewer: 2 Comments for Transmission to the Authors

The manuscript was acceptable in its present forum.

Authors’ Response: We appreciate the reviewer’s comments. Thank you.

Reviewer 3 Report

Review - Manuscript ID: jpm-2256680

   Title:  “Association of Diabetes mellitus with all-cause and cause-specific mortality among patients with metabolic dysfunction as sociated fatty liver disease: A longitudinal cohort study.

Comments: The work is extremely relevant. It allows a comprehensive yet defined view that determines the prognostic implications of DM in patients with different forms of fatty liver disease associated with metabolic dysfunction (MAFLD). The consideration that the clinical focus of MAFLD should change from being a single organ entity to a multisystem disease is supported by the results obtained.

This reviewer makes a single suggestion: Insert images of livers considered to have severe steatosis and non-invasive liver fibrosis (lines 146-147).

Observation:

1. References: standardize the way of writing the name of journals (capital letters only in the first word or in all words); Confirm that the reference Le, M.H.; Yeo, Y.H.; Li, X.; Li, J.; Zou, B.; Wu, Y.; Ye, Q.; Huang, D.Q.; Zhao, C.; Zhang, J.; et al. 2019 Global NAFLD Preva-379 lence: A Systematic Review and Meta-analysis. Clinical gastroenterology and hepatology : the official clinical practice journal of the 380 American Gastroenterological Association 2021, doi:10.1016/j.cgh.2021.12.002, is correct?

Author Response

Reviewer: 3 Comments for Transmission to the Authors

Comments: The work is extremely relevant. It allows a comprehensive yet defined view that determines the prognostic implications of DM in patients with different forms of fatty liver disease associated with metabolic dysfunction (MAFLD). The consideration that the clinical focus of MAFLD should change from being a single organ entity to a multisystem disease is supported by the results obtained.

This reviewer makes a single suggestion: Insert images of livers considered to have severe steatosis and non-invasive liver fibrosis (lines 146-147).

Authors’ Response: We appreciate the reviewer’s comments. The ultrasonic examinations of NHANES III were originally recorded using a Toshiba Sonolayer SSA-90A and Toshiba video recorder. Three ultrasound readers were trained by a board certified radiologist who is specialized in hepatic imaging in order to evaluate hepatic steatosis. Hepatic fibrosis is defined by a non-invasive fibrosis score. Unfortunately, we cannot access to the original examination images, and only the ultrasonic examination results are publicly available. (https://wwwn.cdc.gov/nchs/data/nhanes3/34a/HGUHS.htm)

Observation: 1. References: standardize the way of writing the name of journals (capital letters only in the first word or in all words); Confirm that the reference Le, M.H.; Yeo, Y.H.; Li, X.; Li, J.; Zou, B.; Wu, Y.; Ye, Q.; Huang, D.Q.; Zhao, C.; Zhang, J.; et al. 2019 Global NAFLD Preva-379 lence: A Systematic Review and Meta-analysis. Clinical gastroenterology and hepatology : the official clinical practice journal of the 380 American Gastroenterological Association 2021, doi:10.1016/j.cgh.2021.12.002, is correct?

Authors’ Response: We appreciate the reviewer’s observation. We have revised the format of references according to the reviewer’s comments.